# Minimizing the effects of Pb-loss in detrital and igneous U-Pb zircon geochronology by CA-LA-ICP-MS

Erin E. Donaghy[1], Michael P. Eddy[1], Federico Moreno[2], Mauricio Ibañez-Mejia[2]

[1]Department of Earth, Atmospheric, and Planetary Sciences, Purdue University, West Lafayette, 47907, United States of America

[2]Department of Geosciences, University of Arizona, Tucson, 85721, United States of America

*Correspondence to:* Erin E. Donaghy (edonaghy@purdue.edu)

**Abstract.** Detrital zircon geochronology by laser ablation-inductively coupled plasma-mass spectrometry (LA-ICP-MS) is a widely-used tool for determining maximum depositional ages, sediment provenance, and reconstructing sediment routing pathways. Although the accuracy and precision of U-Pb geochronology measurements has improved over the past two decades, Pb-loss continues to impact the ability to resolve zircon age populations by biasing affected zircon toward younger apparent ages. Chemical abrasion (CA) has been shown to reduce or eliminate the effects of Pb-loss in zircon U-Pb geochronology, but has yet to be widely applied to large-n detrital zircon analyses. Here, we assess the efficacy of the chemical abrasion treatment on zircon prior to analysis by LA-ICP-MS and discuss the advantages and limitations of this technique in relation to detrital zircon geochronology. We show that i) CA does not systematically bias LA-ICP-MS U-Pb dates for thirteen reference materials that span a wide variety of crystallization dates and U concentrations; ii) CA-LA-ICP-MS U-Pb zircon geochronology can reduce, or eliminate, Pb-loss in samples that have experienced significant radiation damage; and iii) bulk CA prior to detrital zircon U-Pb geochronology by LA-ICP-MS improves the resolution of age populations defined by $^{206}Pb/^{238}U$ dates (Neoproterozoic and younger) and increases the percentage of concordant analyses in age populations defined by $^{207}Pb/^{206}Pb$ dates (Mesoproterozoic and older). The selective dissolution of zircon that has experienced high degrees of radiation damage suggests that some detrital zircon age populations could be destroyed or have their abundance significantly modified during this process. However, we did not identify this effect in either of the detrital zircon samples that were analyzed as part of this study. We conclude that pre-treatment of detrital zircon by bulk CA may be useful for applications that require increased resolution of detrital zircon populations and increased confidence that $^{206}Pb/^{238}U$ dates are unaffected by Pb-loss.

## 1. Introduction

Detrital zircon U-Pb geochronology is a widely-used tool with a broad range of applications across multiple subdisciplines of geology. As the efficiency, accuracy, and precision of U-Pb geochronology measurements continue to improve (e.g., Carrapa, 2010; Gehrels, 2012; Gehrels, 2014; Pullen et al., 2014; Sundell et al., 2021), the production of large detrital zircon datasets by laser ablation-inductively coupled plasma-mass spectrometry (LA-ICP-MS) has become more common. In basin analysis and tectonics, these datasets are often used to determine sediment provenance, characterize source terranes, estimate maximum depositional ages, and reconstruct ancient sediment routing pathways (Fedo et al., 2003; Anderson 2005; Smith et al., 2023). The resulting data is typically interpreted using kernel density estimates (KDEs) or probability density plots (PDPs) and assessed by comparing the means,

heights, widths, and modes of peaks in detrital zircon age spectra using similarity/dissimilarity metrics. One factor that may limit the resolution of these peaks is Pb-loss which can smear zircon age populations toward younger apparent U-Pb dates. This issue may not bias data in which Pb-loss is a recent phenomenon provided that the $^{207}$Pb/$^{206}$Pb date is used for zircon crystallization. However, protracted or complicated histories of Pb-loss can make it difficult to interpret $^{207}$Pb/$^{206}$Pb dates (Nemchin and Cawood, 2005) and many labs only use this system to constrain a zircon crystallization date if it is concordant. The precision of the $^{207}$Pb/$^{206}$Pb chronometer also typically limits its use to Mesoproterozoic and older zircon. The most precise date for Neoproterozoic or younger zircon is generally obtained with the $^{206}$Pb/$^{238}$U chronometer and these dates are more susceptible to open-system behavior. Zircon age populations that are affected by Pb-loss in this age range can be difficult to identify since Pb-loss trajectories closely follow Concordia and may result in analyses that are concordant within analytical uncertainty but have spuriously young $^{206}$Pb/$^{238}$U dates. This is especially problematic for the estimation of maximum depositional ages (MDAs) in detrital zircon datasets where age estimations utilize low-n clusters of the youngest zircon ages (Dickinson and Gehrels, 2009; Herriott et al., 2019; Coutts et al., 2019; Sharman et al., 2020; Vermeesch, 2021). The effect of Pb-loss on detrital zircon analyses is consequently two-fold. It reduces the number of concordant Mesoproterozoic and older zircons, making populations in this age range more difficult to identify, and it will cryptically smear Neoproterozoic and Phanerozoic zircon age populations along concordia toward spuriously young dates, making it difficult to resolve differences between distinct but similarly aged populations.

While high temperature metamorphism may lead to zircon recrystallization and partial, or total, resetting of the U-Pb system, most Pb-loss in zircon that is hosted in sedimentary strata represents a low temperature process. Damage to the zircon crystal lattice occurs during each alpha emission along the $^{238}$U, $^{235}$U, and $^{232}$Th decay chains as the heavy nucleus recoils (Bateman, 1910; Dickin, 2005; Nasdala et al., 2005; Reiners, 2005). At temperatures below 200°C this damage cannot anneal and begins to accumulate (Marsellos and Garver, 2010; Ginster et al., 2019). Areas where high levels of damage have accumulated are then susceptible to Pb-loss (Chakoumakos et al., 1987; Mattinson et al., 1994; Garver and Kamp, 2002; Widmann et al., 2019; McKanna et al., 2023). The mechanisms through which Pb becomes mobile in metamict zircon grains remain understudied, but likely include mobility in low-temperature aqueous fluids (Goldich and Mudrey, 1972; Black, 1987; Kramers et al., 2009; Keller et al., 2019), which allows water to penetrate highly radiation damaged zircon and mobilize radiogenic Pb by changing its redox state (Kramers et al., 2009). Thus, the zircons that are most susceptible to Pb-loss at low temperatures are those that spend long durations at shallow crustal levels and encounter low-temperature aqueous fluids, both of which are conditions seen by detrital zircon hosted in sedimentary basins.

The chemical abrasion method, in which thermally annealed zircon is partially dissolved in hydrofluoric acid (HF) prior to analysis has been shown to successfully mitigate low-temperature Pb-loss (e.g., Mundil et al., 2004; Mattinson, 2005; Widmann et al., 2019; Sharman and Malkowski, 2023) and is widely used in isotope dilution-thermal ionization-mass spectrometry (ID-TIMS) U-Pb zircon geochronology (*see reviews in* Schoene, 2014; Schaltegger et al., 2015). The technique likely benefits analyses in two ways. First, it selectively dissolves zones of the zircon crystal that have experienced extensive radiation damage and possible Pb-loss (Widmann et al., 2019; McKanna et al., 2023). Second, the partial dissolution process dissolves inclusions that may harbor non-radiogenic Pb, leading to a higher

proportion of zircon-hosted radiogenic Pb (Pb*) in the measured analysis. Over the last decade, several groups have
analyzed chemically abraded zircon by LA-ICP-MS and shown that this approach can successfully mitigate Pb-loss,
and results in increased concordance, precision, and, presumably, accuracy of U-Pb dates (Crowley et al., 2014; Von
Quadt et al., 2014). These results suggest that chemical abrasion prior to large-n detrital zircon analyses may also be
useful when the resolution of closely spaced Neoproterozoic and Phanerozoic peak age populations is desired or when
high degrees of discordance obscure the interpretation of Mesoproterozoic and older age populations. Here, we assess
the benefits and drawbacks of this pre-treatment with a particular focus on whether the resolution of younger zircon
age populations is increased, whether it improves concordance for Precambrian detrital zircon populations, and/or
whether the selective removal of metamict zircon will bias age populations.

**2. U-Pb Zircon Geochronology Approach and Methods**
We have divided our study into three distinct parts. First, we compare chemically abraded and untreated
zircon from 13 zircon reference materials (Table 1) to test whether chemical abrasion systematically biases U-Pb dates
analyzed by LA-ICP-MS. Crowley et al. (2014) demonstrated that chemically abraded zircon ablate more slowly and
experience greater down-hole fractionation than untreated zircon. These differences are likely related to changes in
the ability of the laser to couple with zircon that has been etched by the chemical abrasion process. While no negative
effects of chemical abrasion were seen in Crowley et al. (2014) or von Quadt et al. (2014), provided that chemically
abraded reference materials were used for instrument calibration, we have expanded the age range of reference zircon
analyzed to encompass 28.5 – 3467 Ma. This increased age range of the tested reference materials provides a more
complete understanding of LA-ICP-MS U-Pb systematics on chemically abraded zircon and whether a single primary
reference material can be used to calibrate the instrument for a wide range of zircon dates. Second, we assess the
ability of chemical abrasion to mitigate Pb-loss in an igneous sample that has experienced substantial radiation damage
by comparing chemically abraded and non-chemically abraded $^{206}Pb/^{238}U$ LA-ICP-MS zircon analyses to a newly
produced CA-ID-TIMS reference date for a Mesoproterozoic granite. Finally, we assess how CA affects detrital zircon
(DZ) age spectra by comparing chemically abraded and untreated aliquots of two detrital samples. One sample is
Cenozoic in age and contains both Phanerozoic (100-300 Ma) and Precambrian (1000-1200 Ma) zircon age
populations, whereas the second sample is Proterozoic and contains zircon age populations between 2000-3500 Ma.

**2.1 Methods for Thermal Annealing and Chemical Abrasion**
All chemically abraded zircon aliquots were treated at Purdue University following methods modified from
Mattinson (2005) and similar to those described in Eddy et al. (2019). Zircon separates were first thermally annealed
in quartz crucibles for 60 hours at 900°C in a muffle furnace and then loaded in 3 mL savillex hex beakers with ~1
mL of 28M HF and 0.1 mL of 8M HNO$_3$ for bulk chemical abrasion. Four hex beakers were then stacked in the PTFE
liner for a 125 mL Parr acid dissolution vessel. To ensure vapor exchange during partial dissolution a small hole was
drilled through each beaker cap. The fully assembled Parr acid dissolution vessel was then held at 210°C for 12 hours.
Once the chemical abrasion process was completed, the leachate was removed from each beaker using a pipette and
the zircons were rinsed three times with H$_2$O. Chemically abraded aliquots were then sent to the University of Arizona
LaserChron Center (ALC) for mounting and LA-ICP-MS analyses. Methods for chemical abrasion of zircon prior to
the ID-TIMS analyses reported in this paper are similar to those described above, except individual zircon were
chemically abraded in 200 µL Ludwig style microcapsules and repeatedly rinsed in distilled 7M HCl and ultrapure
$H_2O$ prior to spiking and complete dissolution.

**Table 1.** Zircon reference materials for U-Pb isotopic analyses

| Name | ID-TIMS age (Ma) | $2\sigma$ | References | Host lithology | Quantity |
|---|---|---|---|---|---|
| Fish Canyon Tuff | 28.476 | 0.029 | Schmitz and Bowring (2001)[b, c] | Dacite | Unlimited |
| GHR1 | 48.106 | 0.023 | Eddy et al. (2019)[b] | Rapakivi Granite | Unlimited |
| 49127 | 136.6 | | Gehrels et al. (2008)[b] | | Uncertain |
| Plesovice | 337.13 | 0.37 | Slama et al. (2008)[a] | Potassic Granulite | Unlimited |
| Temora 2 | 418.37 | 0.14 | Mattinson (2010)[a] | Gabbro | Unlimited |
| R33 | 420.53 | 0.16 | Mattinson (2010)[a] | Monzodiorite | Unlimited |
| SLM | 563.5 | 3.2 | Gehrels et al. (2008)[b] | Single Crystal | Limited |
| SLF | 555.86 | 0.68 | Wang et al. (2022)[b] | Single Crystal | Limited |
| 91500 | 1065.4 | 0.3 | Wiedenbeck et al. (1995)[b] | Single Crystal | Limited |
| FC1 | 1098.47 | 0.16 | Mattinson (2010)[a] | Gabbro | Unlimited |
| Oracle | 1434 | 8 | Gehrels et al. (2008)[b] | Granite | Unlimited |
| QGNG | 1851.6 | 0.6 | Black et al. (2004)[b] | Quartz gabbro gneiss | Uncertain |
| OG1 | 3467.05 | 0.63 | Stern et al. (2009)[a] | Diorite | Unlimited |

[a] Chemical abrasion CA-ID-TIMS
[b] Traditional ID-TIMS
[c] CA-ID-TIMS analyses by Wotzlaw et al. (2013) show significant age dispersion in Fish Canyon Tuff relative to original U-Pb ID-TIMS date of Schmitz and Bowring (2001).


### 2.2 LA-ICP-MS Zircon U-Pb Geochronology

Zircon aliquots were mounted in 2.5-cm-diameter epoxy plugs, polished, and imaged by
cathodoluminescence using a Hitachi 3400N SEM and a Gatan Chroma CL system prior to analysis by LA-ICP-MS.
Chemically abraded zircon were only mounted with chemically abraded zircon reference materials, while untreated
zircon aliquots were mounted with untreated reference materials. U-Pb isotopic analyses were obtained via LA-ICP-
MS using a Thermo Element2 single-collector ICP-MS coupled with a Teledyne Photon Machines Analyte G2
excimer laser at the ALC. The diameter of the laser spot was set to 30 microns. Elemental- and mass-dependent
instrumental fractionation were corrected by bracketing unknown analyses with analyses of primary reference material
FC1 following the methods described in Pullen et al. (2018). Please see supplementary Table S23 for tuning
parameters for the laser and mass spectrometer. Only chemically abraded primary reference materials were used for
calibration of chemically abraded samples and only untreated primary reference materials were used for untreated
samples following the recommendations of Crowley et al. (2014). Bracketing of secondary and tertiary reference
materials occurred every 10-11 analyses with primary reference materials (FC1, SLF/SLM, R33) for the round-robin
comparison of zircon reference materials, every 2-3 analyses for igneous zircon analyses, and every 5 analyses for
detrital zircon samples. Data reduction was completed using an in-house Matlab script, AgeCalcML v.1.42 (Sundell
et al., 2021). This program allows the user to filter data by maximum $^{206}Pb/^{238}U$ and/or $^{207}Pb/^{206}Pb$ uncertainty
(typically set to 10%), reverse discordance (typically 5%), and normal discordance (typically 20%). For the purposes
of this study, we de-activated all uncertainty and discordance filters in AgeCalcML and all isotopic data measured via
LA-ICP-MS that is clearly from ablated zircon are reported in Tables S1-S13. However, age interpretations of igneous
and detrital zircon data use filtered data (Tables S16-S21).

141       Uranium concentrations (ppm) reported from routine U-Pb LA-ICP-MS zircon analyses at ALC are

semiquantitative and calibrated by bracketing unknowns with analyses of reference materials with a known average
U concentration. However, since chemical abrasion selectively dissolves high-uranium zones, and thus modifies the
average U concentration of reference materials by an unquantified amount, the reported U concentration values for
reference materials analyzed by traditional (non-CA) ID-TIMS may no longer be valid. We found that the primary
reference material SLF had homogenous $^{238}$U cps (counts per second) within individual sessions (Figs. S15-S17) for
both the CA and non-CA runs. We interpret this to mean that SLF has a homogenous U concentration (and radiation
damage) and that any differences in $^{238}$U cps for SLF between different analytical sessions are related to changes in
the instrument's sensitivity. As such, we normalize the $^{238}$U cps for all other grains analyzed in each session to the
average $^{238}$U cps of SLF across all sessions. Given the potential difference between chemically abraded and untreated
SLF U concentrations, we did this correction independently for treated and untreated grains. Since we do not have U
concentration values for treated and untreated SLF, we center our discussions on relative differences in U
concentration as estimated using the intensity (in cps) of the $^{238}$U beam, rather than quantifying U concentrations.

154       We used Saylor and Sundell (2016) DZstats program to complete a quantitative assessment of the similarity

between treated and untreated aliquots. This program implements five tests to compare large-n geochronologic or
thermochronologic datasets. The tests used in this study are similarity, likeness, and cross-correlation. Results from
all five tests are shown in Supplemental Table S22 for both detrital zircon samples. The similarity coefficient measures
if two samples have similar modal sub-intervals as well as similar proportions of components in each one of those
modes. A value for similarity equal to 1 indicates the samples are identical in both peak modes and proportions,
whereas 0 indicates there is no match between modes and proportions (Saylor and Sundell, 2016). This test is useful
in assessing the number of peak age populations (similar mode intervals) and how peak heights (proportion of
components in each mode) change between two samples. The cross-correlation coefficient is also sensitive to the
presence or absence of peak ages, but also changes due to the relative magnitude and shape of peaks. If a sample
shared the same peak ages, peak shapes, and magnitude of peaks, it would have a $R^2$ value of 1. If no peak ages, peak
shapes, and magnitude of peaks are shared, the $R^2$ value would be 0 (Saylor and Sundell, 2016). Likeness is the
complement of the area of mismatch between two detrital zircon spectra, or more simply put the degree of "sameness"
between detrital zircon age populations (Satkoski et al., 2013). Thus, the likeness test compares the degree of overlap
between pairs of PDPs and is a measure of resemblance between proportions of two populations with overlapping
ages (Gehrels, 2009; Satkoski et al., 2013). Values of likeness that approach 1 indicate that two detrital zircon spectra
have a high degree of overlap (Satkoski et al., 2013; Saylor and Sundell, 2016).



 **2.3 Zircon Optical Profilometry**

To evaluate the effect of CA on laser ablation excavation rates in zircons, we compared the average depth at
increasing ablation times on a series of laser pits on treated and untreated reference materials. This was accomplished
by generating ten laser ablation pits with excavation times that increased by three seconds (starting at three and
increasing to thirty seconds) in single crystals of treated and untreated grains of zircon reference materials FC1, R33,
and SL. The resulting pits were imaged using a Veeco Wyko NT9800 Optical Profilometer and depth maps, 3-D
images, and crosscut profiles were created using the Vison software produced by Veeco. The images and profiles
allowed for the estimation of pit depths and can be used to calculate excavation rates when combined with the known
ablation periods (Fig. 2). Laser ablation pits were also imaged and measured on three treated and five untreated
unknowns from sample MIGU-02.

**2.4 CA-ID-TIMS Zircon U-Pb Geochronology**
Sample MIGU-02, a granitoid from the Guyana Shield, was analyzed by CA-ID-TIMS at Purdue University
to provide a reference date to compare the chemically abraded and untreated LA-ICP-MS analyses. Following the
chemical abrasion methods described above, individual zircons were spiked with the EARTHTIME $^{205}$Pb-$^{233}$U-$^{235}$U
isotopic tracer (Condon et al., 2015; McLean et al., 2015) and loaded into a Parr acid digestion vessel with 28M HF.
The vessel was then held at 210°C for 60 hours for zircon dissolution. After dissolution, the samples were dried down
and converted to chloride form, by adding 75 μl 7M HCl, reassembling the Parr acid digestion vessel, and holding it
at 180°C for 12 hours. After conversion to chloride form, the solution was converted to 3M HCl in preparation for
anion exchange chromatography. Pb and U were purified from these solutions using AG-1x8 anion exchange resin
following procedures modified from Krogh (1973). The resulting aliquots were dried down to a chloride salt before
being dissolved in silica gel, dried onto rhenium filaments, and loaded into an IsotopX Phoenix TIMS for analysis. Pb
isotopic measurements were made by peak hopping on a Daly detector and corrected for mass dependent isotopic
fractionation using an α= 0.147 ± 0.028 (%amu) and deadtime = 29.9 ns, derived from repeat measurements of the
NBS981 Pb reference material. We assume that all $^{204}$Pb is from laboratory contamination and correct for it using a
laboratory Pb isotopic composition of $^{206}$Pb/$^{204}$Pb = 18.82 ± 0.74 (2σ), $^{207}$Pb/$^{204}$Pb = 15.52 ± 0.63 (2σ), $^{208}$Pb/$^{204}$Pb =
37.93 ± 1.60 (2σ) derived from repeat total procedural blank measurements run during 2022. Uranium was run as an
oxide (UO$_2$) and  isotopic measurements were made statically using Faraday detectors and corrected for fractionation
using the known ratio of $^{233}$U/$^{235}$U in the EARTHTIME $^{205}$Pb-$^{233}$U-$^{235}$U isotopic tracer (Condon et al., 2015; McLean
et al., 2015) and assuming a zircon $^{238}$U/$^{235}$U value of 137.818 ± 0.045 (Hiess et al., 2012). Data reduction was done
using the ET_Redux software package (Bowring et al., 2011) and the decay constants of Jaffey et al. (1971). All
isotopic data measured via CA-ID-TIMS are presented in Supplemental Table S17.

**3. Results**
**3.1 CA-LA-ICP-MS U-Pb Geochronology of Zircon Reference Materials**
Treated and untreated aliquots of thirteen different zircon U-Pb reference materials (Table 1) were analyzed
in this study to further assess whether chemical abrasion systematically biases U-Pb dates. The reference materials
were analyzed during two round-robin runs using the approach described above. The first run targeted 15 zircon grains
from treated and untreated aliquot of reference materials. During the second run, 30 zircon grains were targeted.
Because FC1 was used as a primary reference material for calibration of the LA-ICP-MS, approximately 30 FC-1
zircons were analyzed during run one and 87 were analyzed during run two in both treated and untreated aliquots.
This led to 117 FC1 grains analyzed per treated and untreated aliquots of reference material. The total number of
zircons analyzed was 653 in each of the chemically abraded and untreated aliquots of reference materials. Of the 653
grains in the chemically abraded aliquots, 631 analyses (96.6%) were retained following filtering for discordance,
whereas 608 analyses (93.5%) were retained in the untreated aliquot. These results further confirm that CA helps
mitigate Pb-loss and improve the percentage of retained concordant LA-ICP-MS analyses (e.g., Crowley et al., 2014;
von Quadt et al., 2014). The most extreme change in concordance and data retention occurred between treated and
untreated FC-1 zircon (1098.4 Ma). Of the 117 grains analyzed in both the treated and untreated aliquots, 99.1% of
analyses were retained in the chemically abraded aliquot versus 82.2% in the untreated aliquot. Discordance criteria
used for filtering the above data were reverse discordance larger than 5%, $^{206}Pb/^{238}U$ errors larger than 10%, and/or
maximum discordance of over 20%. Overall, the discordant FC1 grains in both runs had low U cps and significantly
older $^{206}Pb/^{238}U$ dates (>1250 Ma; Fig. S15).
The weighted mean dates of CA and non-CA reference materials are all within $0.1 - 4\%$ of the reference
ages determined by ID-TIMS (Fig. 1). Therefore, despite an increase in the percent of concordant treated grains
relative to untreated grains, weighted means of acceptable analyses are indistinguishable and indicate that it is unlikely
that chemical abrasion biases U-Pb dates within LA-ICP-MS instrument uncertainty. The greatest scatter in calculated
weighted mean ages (~4 to 0.1% age offset from reference date) is in both the treated and untreated Mesozoic to
Cenozoic reference materials. Scatter is improved by chemical abrasion in Paleozoic reference materials (2 to 0.8%
age offset) and excellent for Proterozoic and Archean aliquots (0.6 to -0.7%). Additionally, concordant analyses of
treated aliquots have overall lower $^{238}U$ cps compared to the untreated aliquots (Fig. S15), indicating that chemical
abrasion dissolved zones of high U concentrations where Pb-loss is most likely to have occurred (Widmann et al.,
2019; McKanna et al., 2023). Since reference materials are selected for their homogeneous isotopic compositions, it
is not surprising that there is similarity in dates between treated and untreated aliquots. The reproducibility of U-Pb
dates for all of the reference materials is strong evidence that a single primary reference material (FC-1 in this case)
can be used to correct for instrumental fractionation across a wide range of zircon ages and trace element compositions
for chemically abraded zircon.
Despite the overall similarity in bias between treated and untreated reference materials, the behavior of some
reference materials warrants further discussion. The CA-LA-ICP-MS weighted mean $^{206}Pb/^{238}U$ dates for two
Cenozoic reference materials were older than the CA-ID-TIMS reference date. Chemical abrasion of GHR1 zircon
led to an increase of concordant grains, but an older $^{206}Pb/^{238}U$ weighted mean date (Fig. S2). We attribute this
difference to the presence of slightly older xenocrysts within the sample (e.g., Eddy et al., 2019). We see a similar
result for Fish Canyon tuff zircon where the CA aliquot showed increased concordance, but the calculated mean age
was offset more from the reference age than the no-CA aliquot (Fig. S1). This sample contains significant antecrysts
that might bias its results (e.g., Wotzlaw et al., 2013). Indeed, increased precision and accuracy in analyses of young
suites of igneous zircon routinely find overdispersion that can be related to protracted zircon growth or the presence

of xenocrysts/antecrysts. Thus, the slight variability in weighted mean dates for GHR1 and Fish Canyon samples in CA-LA-ICP-MS analyses is not entirely unexpected and therefore unlikely to reflect of a systematic bias of the CA-LA-ICP-MS method. Additionally, the 91500 reference material has shown substantial negative age offset in other studies (Gehrels et al., 2008; Schoene et al., 2014), but the origin of these offsets has remained enigmatic, and the offset in this study is not surprising.

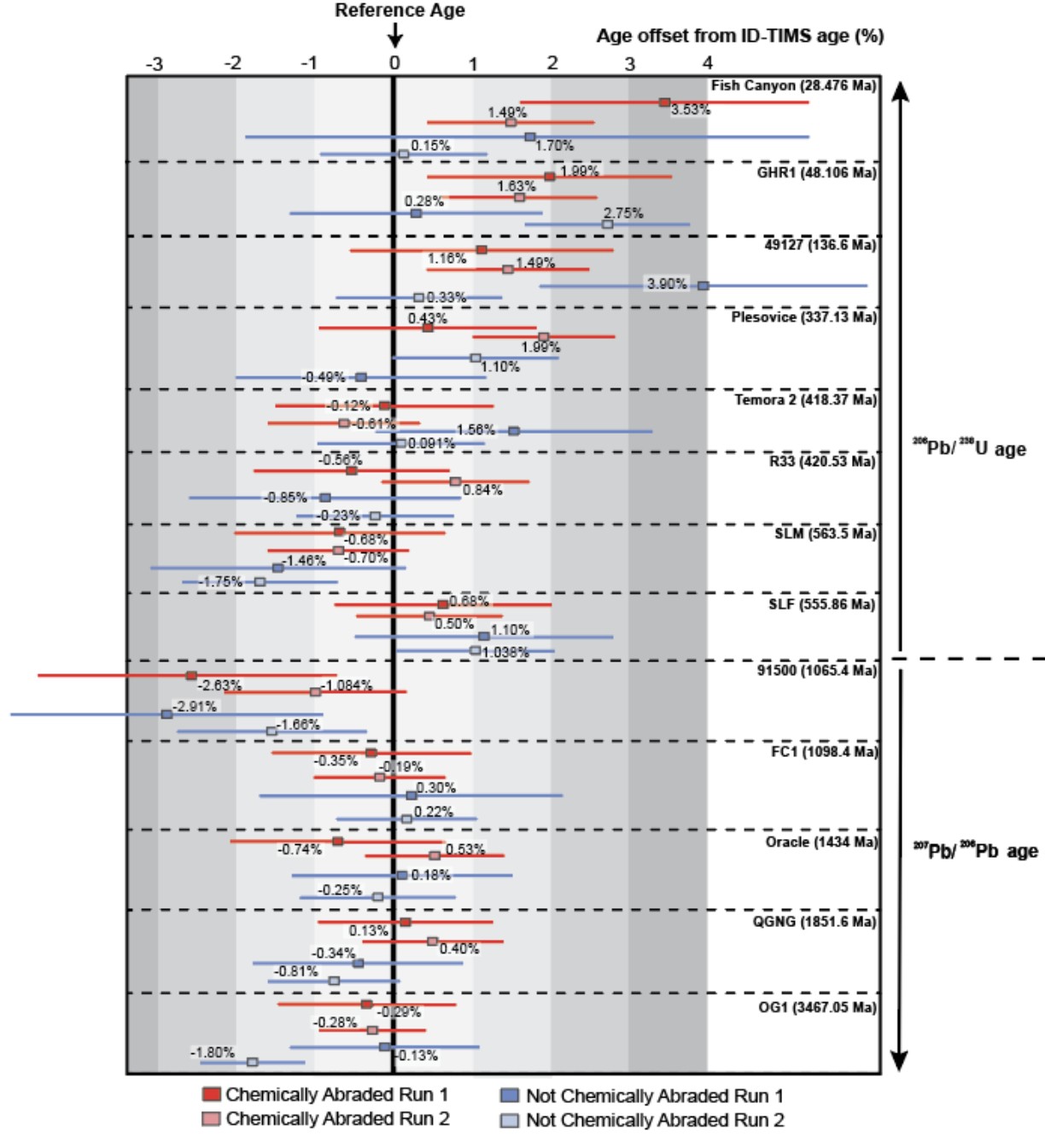

**Figure 1.** Comparison of $^{206}Pb/^{238}U$ and $^{207}Pb/^{206}Pb$ (CA)-LA-ICP-MS ages with CA-ID-TIMS ages for thirteen reference materials that range in age from 28 to 3467 Ma. Each square is the weighted mean of a set of (CA)-LA-ICP-MS measurements shown as the percent offset from the known reference age (ID-TIMS). The uncertainty is reported as 2-sigma standard error of the weighted mean. Chemical abrasion of treated aliquots was conducted at Purdue University and laser ablation analyses were conducted at Arizona LaserChron Center on the Thermo Element2 single-collector ICP-MS. Methods for LA-ICP-MS at LaserChron using the Element2 are described by Pullen et al. (2018).

Variations in laser ablation behavior between primary reference materials used for standardization and samples is a direct result of differences in zircon matrices and are known as 'matrix effects' (Marillo-Sialer et al., 2016). Differences in zircon matrices are related to numerous factors (see review in Marillo-Sialer et al., 2016), including presence/absence of trace elements (Black et al., 2004), the amount of radiation damage (Allen and Campbell, 2012), and the degree of crystallinity (Steely et al., 2014). These factors all impact the laser's ability to couple with the surface of the zircon, directly impacting laser ablation rates and the rates of down-hole fractionation. As a result, matrix-effects can lead to systematic shifts in LA-ICP-MS data and may be a contributor in observed shifts in treated and untreated aliquots of reference materials in this study (Fig. 1). In order to constrain how CA impacts laser coupling and ablation rates between treated and untreated reference materials, we ablated 10 spots on a single treated and untreated zircon crystal of FC1, SL, and R33 (n=60; Table S14). For each individual spot 1-10 on a reference material, we incrementally raised the ablation duration by 3 seconds (i.e., by 21 individual laser pulses at 7 Hz). This resulted in 10 different spots with ablation durations ranging from 3 to 30 seconds (21 to 210 laser pulses) and allowed us to calculate laser ablation rates for both treated and untreated zircon. Overall, laser ablation rates varied linearly with time (0.43 to 0.46 um/sec) and were similar for treated and untreated aliquots of all primary reference materials (Fig. 2). Similar ablation rates were observed across all three different treated and untreated aliquots of primary reference materials. This suggests 1) the primary reference materials have similar zircon matrix densities and ablate at similar rates (Marillo-Sialer et al., 2014; 2016), and 2) chemical abrasion does not change the zircon matrix density or alter the zircon surface of primary reference materials in a way that drastically alters laser coupling or ablation rates.

283

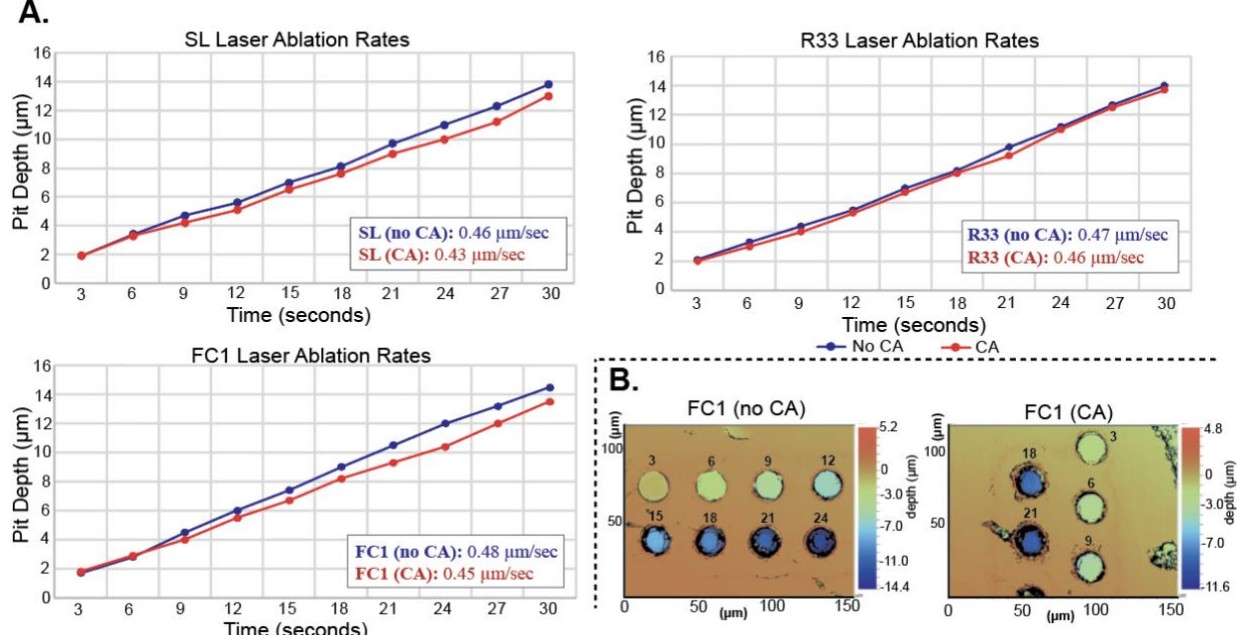

**Figure 2. A**. Line graphs showing pit depths (µm) versus time (seconds) for unabraded and abraded aliquots of primary reference materials. Each data point represents an individual pit. Untreated reference materials have slightly deeper pit depths and ablate at marginally faster rates, but overall, rates of ablation for treated and untreated reference materials are similar. **B.** An example of one of the VEECO surface data maps showing pit depths on treated and untreated zircon of FC1. Pits are labeled with the duration in seconds of ablation. See Supplemental Table S14 for all pit depth data from abraded and unabraded reference materials.

### 3.2 Untreated and CA- U-Pb Zircon LA-ICP-MS Analyses of Metamict Zircon

A Precambrian granite sample from the Parguaza Complex in the North Guyana Shield (MIGU-02; N 5° 21' 3.70"; W 67° 41' 33.41") that has experienced substantial radiation damage was analyzed to assess the effects of chemical abrasion on grains with significant Pb-loss. Untreated (n = 35) and treated aliquots (n = 23) of MIGU-02 were analyzed at the ALC and compared to a reference age determined by CA-ID-TIMS (n=6) at Purdue University (Fig. 3; Tables S14 and S15). During the bulk chemical abrasion process, 80-85% of MIGU-02 grains fully dissolved, leaving only a small fraction of the original aliquot to be used for analyses.

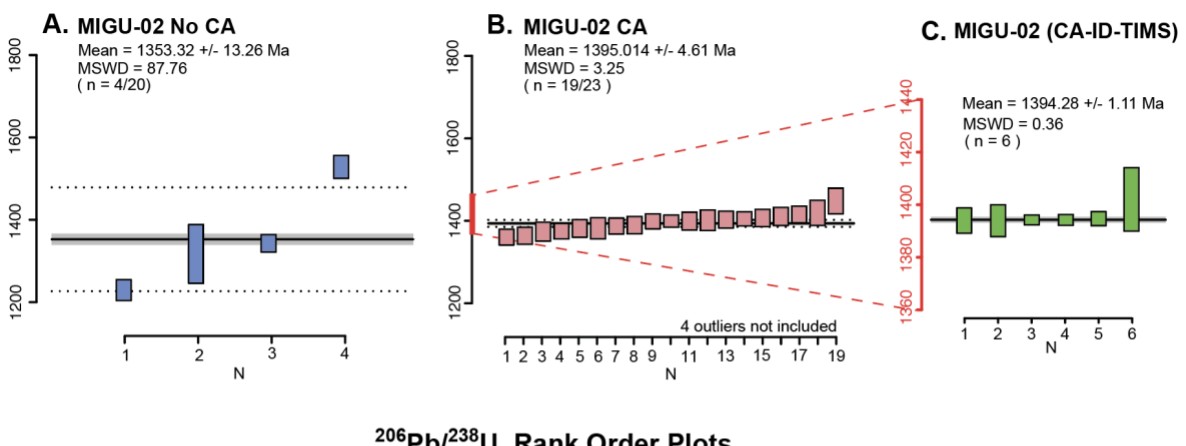

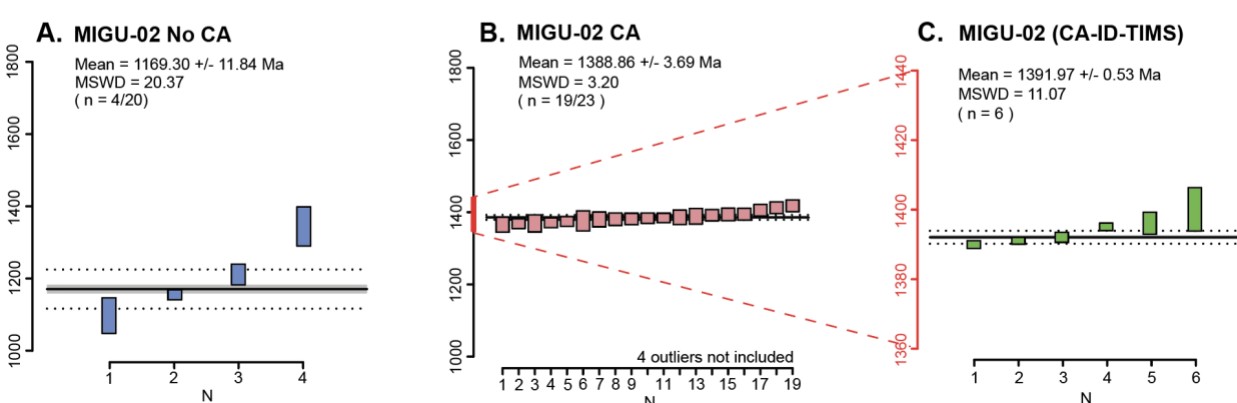

**Figure 3.** Rank order plots of calculated $^{207}Pb/^{206}Pb$ and $^{206}Pb/^{238}U$ ages for treated and untreated MIGU-02 aliquots and of the reference age for MIGU-02 obtained using CA-ID-TIMS. **A.** Untreated samples of MIGU-02 show large degree of scatter in dates and substantial deviation from the reference age. **B.** Treated zircons show a significant increase in precision and accuracy of ages relative to the reference age. **C.** Reference age for MIGU-02 determined using the weighted mean of six grains. See text for discordance criteria.

The $^{207}Pb/^{206}Pb$ CA-ID-TIMS reference age for MIGU-02 is 1394.28 +/- 1.11 Ma (n=6, MSWD = 0.36), while the $^{206}Pb/^{238}U$ dates are more scattered (Fig. 3). The scatter indicates that U/Pb elemental fractionation occurred during chemical abrasion in one analysis (slight reverse discordance) and residual Pb-loss remained in others (normal discordance)(Fig. 4). Nevertheless, a weighted mean date of the $^{206}Pb/^{238}U$ CA-ID-TIMS dates is 1391.97+/- 0.53 Ma (n = 6, MSWD = 11.06) and indicates that residual Pb-loss only affects the dates at the <0.5% level. Untreated LA-ICP-MS analyses of MIGU-02 show significant discordance (Fig. 4) and only 4 analyses (n=4/20; 80% discordant) were retained after filtering by AgeCalcML v.1.42. Chemical abrasion substantially increased the number of concordant analyses (n = 23/23). Fifteen analyses were removed from the untreated aliquot dataset and seven analyses were removed from the treated aliquot dataset because they hit epoxy and are not included in the totals. Although all grains were concordant in the treated aliquot, four grains were not included in the weighted mean because they had a significantly older $^{207}Pb/^{206}Pb$ dates (1571-1900 Ma) than the CA-ID-TIMS reference date (Table S14) and are likely

xenocrystic. The weighted mean $^{207}$Pb/$^{206}$Pb date from the untreated MIGU-02 aliquot is 1353.32 +/- 13.26 Ma (n =
4/20; MSWD = 87.76) and the treated aliquot is 1395.014 +/- 4.61 Ma (n = 19/23; MSWD = 3.25). The mean $^{206}$Pb/$^{238}$U
date of the untreated aliquot is 1169.30 +/- 11.84 Ma (MSWD = 20.37) and the mean $^{206}$Pb/$^{238}$U date of the treated
aliquot is 1388.86 +/- 3.69 Ma (MSWD = 3.20). Thus, the dates from treated zircon show a significant increase in
concordance, precision, and accuracy relative to the reference date as determined by CA-ID-TIMS (Fig. 3).

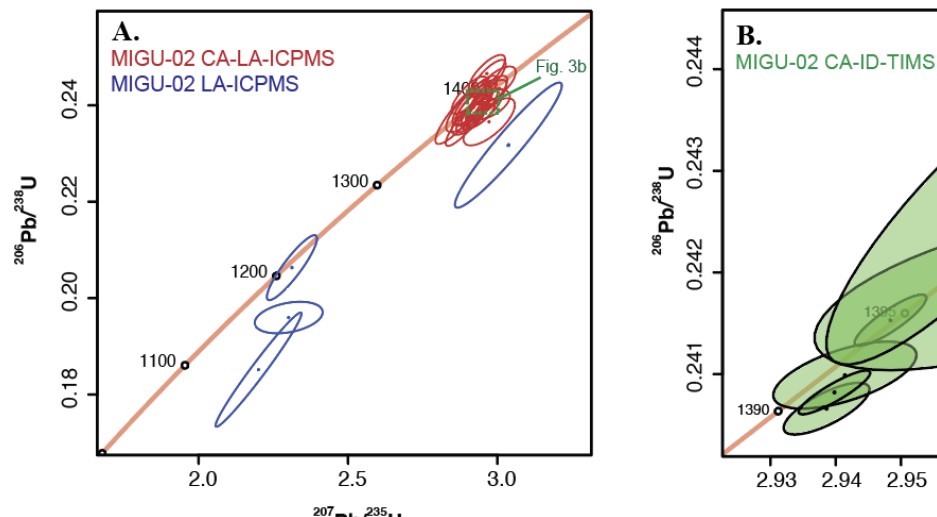


**Figure 4. A.** Untreated and treated aliquots of MIGU-02 shown on a concordia plot. Non-CA MIGU-02 dates are
discordant whereas CA dates fall on concordia and overlap the reference age. **B.** All CA-ID-TIMS analyses of MIGU-
02 shown on a concordia plot. One date shows slight reverse discordance whereas all other dates fall on concordia or
have slight normal discordance.

All the CA-treated analyses have lower $^{238}$U beam intensity than the untreated grains suggesting that CA
selectively removed zones of higher U concentration (Fig. 5). Furthermore, the untreated zircon with the highest $^{238}$U
beam intensity are associated with U-Pb dates that are >±20% discordant (Table S16). Since uranium concentration
is correlated to radiation damage in old zircon, this result reinforces the observation that CA is an effective tool for
removing damaged zones of the zircon (Nasdala et al., 2005; Widmann et al., 2019). Pit depths were measured on five
untreated and three treated zircons from MIGU-02 (Table S15). The average pit depth for untreated grains of MIGU-
02 is ~10.34 µm, whereas it is ~8.1 µm for the treated aliquot. This indicates that the pit depths of the untreated aliquot
of MIGU-02 are ~25% deeper than the treated aliquot and that CA does has an impact on the laser ablation rate in
highly metamict samples.
We acknowledge that for samples with significant radiation damage, there is always the possibility that the
entire sample will dissolve during chemical abrasion, and it is up to the researcher to determine if this technique is
appropriate for their objectives. Running a high-n data acquisition on highly damaged zircon might ultimately yield
enough concordant analyses to make a confident age determination, but analyses of MIGU-02 that passed typical
discordance filters were inaccurate by up to -11% for the $^{207}$Pb/$^{206}$Pb dates and -21% for the $^{206}$Pb/$^{238}$U dates (Fig. 3),
suggesting that even filtered data may be inaccurate for metamict zircon. In contrast, the chemically abraded aliquot
did not have these issues, despite the significant loss of zircon grains during the HF chemical attack.

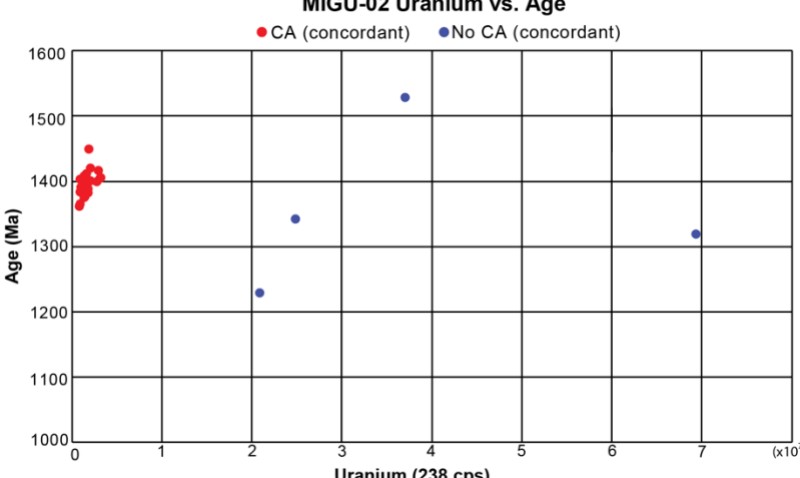

**Figure 5.** Uranium (238 cps) plotted against $^{207}$Pb/$^{206}$Pb age (Ma) for both treated and untreated aliquots of MIGU-02. Only concordant analyses are shown because discordant analyses for MIGU-02 have extremely low $^{238}$U cps. Uranium concentration is directly proportional with radiation damage in zircon with the same low-temperature

cooling history. The restricted range of low $^{238}$U cps in the CA-treated grains suggests that CA was effective at
dissolving high U zircon that was more likely to have Pb-loss.

***3.3 Untreated and CA- U-Pb Zircon LA-ICP-MS Analyses of Detrital Zircon***
One Phanerozoic (NM8A) and one Precambrian sample (Rora Med) were analyzed to determine how detrital
zircon age distributions compare between chemically abraded and untreated aliquots. We followed the 'Large-n'
approach of Pullen et al. (2014) for both treated and untreated aliquots to obtain a more robust distribution of ages,
their modes, peak widths, and abundances. For NM8A, we analyzed 512 individual zircons in the treated aliquot and
896 zircons in the untreated aliquot. In Rora Med, we analyzed 1035 zircons in the treated aliquot and 920 zircons in
the untreated aliquot. We used Saylor and Sundell (2016) DZstats program to complete a quantitative assessment of
the similarity between treated and untreated aliquots (See definitions described above; Table S22). Additionally, we
calculated the fraction of grains that defined distinct peak age populations in each aliquot to assess how the proportion
of each population changed after chemical abrasion. Our results show that chemical abrasion (CA) changed the number
and distribution of apparent peak age populations in both DZ samples compared to the non-CA age spectra (Figs. 6
and 7). Most notably, the Phanerozoic age peaks in sample NM8A narrowed, became more defined, and, in some
cases, shifted to slightly older dates.

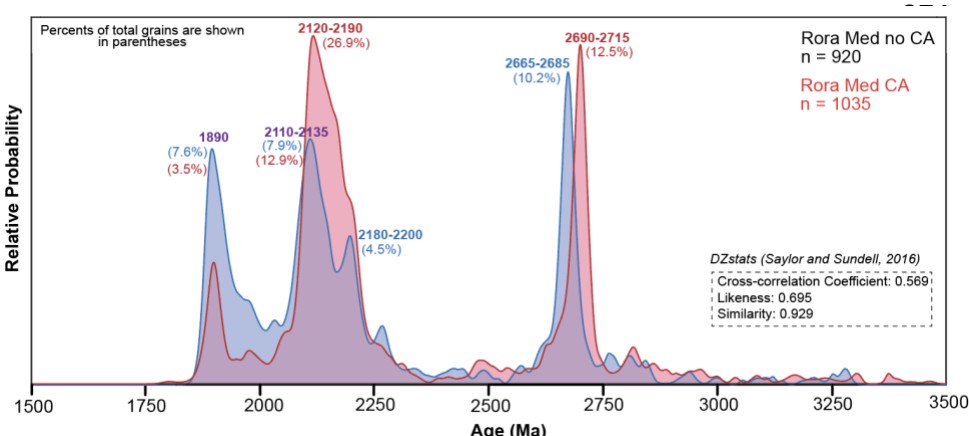

**Figure 6.** Comparison of U-Pb detrital zircon age spectra of not chemically abraded (blue) and chemically abraded (red) aliquots of Rora Med. Areas where age spectra overlap are shaded in purple. Quantitative comparisons of treated and untreated aliquots were completed using Saylor and Sundell (2016) DZstats to calculate cross-correlation, likeness, and similarity values. All quantitative comparison analytics are reported in Table S20. The percentages above the peaks represent the percentage the peak age population represents out of the total grains. We aimed for n=1000 for each aliquot because Pullen et al (2014) shows the distribution of analyzed zircon ages is thought to approach the 'true' age distribution of the sample.

In the Precambrian sample (Rora Med), there are subtle changes in the DZ age spectra between the treated and untreated aliquots. Overall, the CA treated aliquot shows a higher percentage of concordant grains (Fig. 6) narrower, better defined, peak age populations, changes in the number of peaks present, and a slight but noticeable shift in peak age populations to older ages (Fig. 6). Of note, the 1890 Ma peak narrows in the treated aliquot compared to the broad peak that covers a range of ages between 1890 and 2000 Ma in the untreated aliquot. However, the fraction of grains within this population decreased from 7.6% to 3.5% between the untreated and treated fractions, respectively. There is also a change in the shape and number of peaks between the two fractions between 2100 and 2300 Ma. In the untreated aliquot, there are three distinct peak age populations (~2120 (~7.9%), 2190 (~4.5%), & 2260 Ma (<2%)), whereas in the treated aliquot, there is only one broad peak age population that spans between ~2120-2190 Ma (~30%). There is also a distinct shift in the untreated aliquot 2675 Ma peak age population to fifteen million years older in the treated aliquot (Fig. 6). Quantitative comparisons support these observations. A likeness coefficient of 0.695 and cross-correlation coefficient of 0.569 support there are significant differences in the number, shape, and magnitude of certain peak age populations. However, a similarity value of 0.929 indicates that even with these changes for specific age populations, there is an overall high degree of overlap in the number and proportion of modes in the detrital zircon spectra between treated and untreated aliquots.

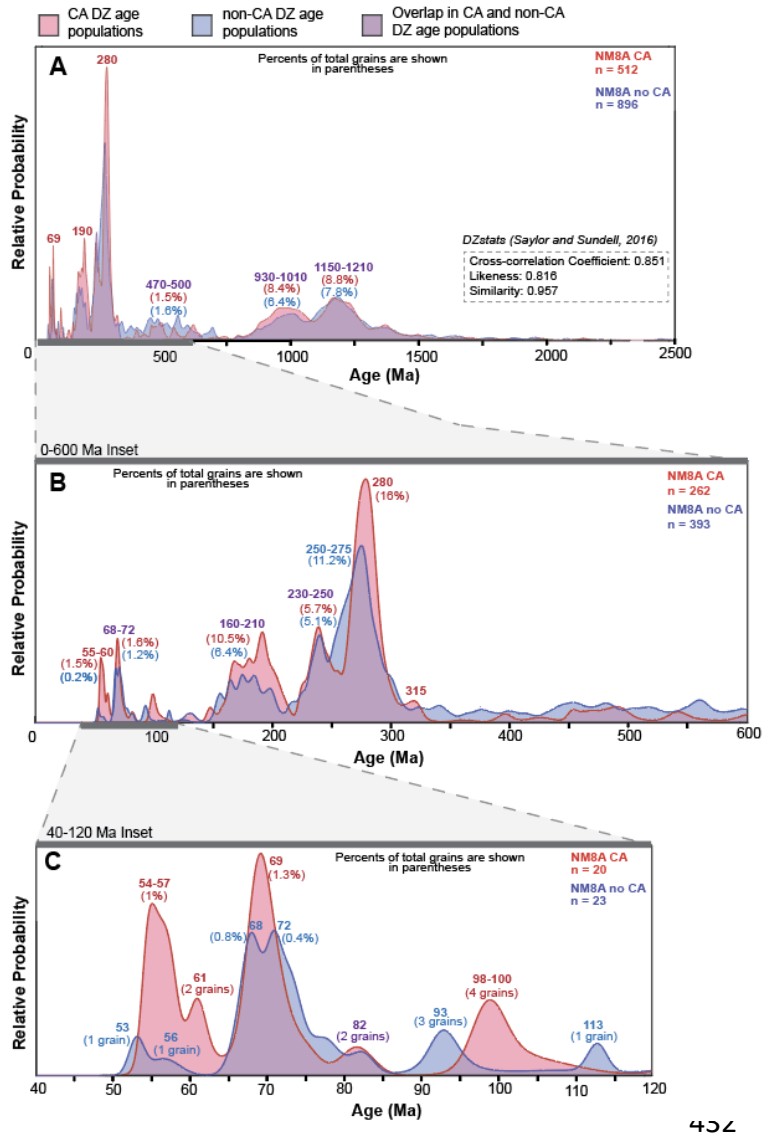

**Figure 7.** Comparison of U-Pb detrital age spectra of not chemically abraded (blue) and chemically abraded (red) aliquots of NM8A. Areas where age spectra overlap are shaded in purple. Quantitative comparisons of treated and untreated aliquots were completed using Saylor and Sundell (2016) DZstats to calculate cross-correlation, likeness, and similarity values. All quantitative comparison analytics are reported in Table S20. The percentages above the peaks represent the percentage the peak age population represents out of the total grains. We aimed for n=1000 for each aliquot as the distribution of analyzed zircons ages is thought to approach the 'true' age distribution of the sample (Pullen et al., 2014). Insets A-C show variations of the scale on the x-axis.

There are also subtle changes in the number of peaks and peak shapes between the treated and untreated aliquots of NM8A. The most significant changes observed are increased resolution and definition of Phanerozoic peak age populations in the treated aliquot (Fig. 7). For example, between 200-300 Ma, two broad peaks in the untreated aliquot sharpen and narrow to two well-defined peak age populations in the treated aliquot (Fig. 7b). Additionally, a distinct 190 Ma peak is present in the treated aliquot compared to a broad range of populations between 160 and 210 Ma in the untreated aliquot. We also see a zone of two broadly defined peaks at 68 and 72 Ma in the untreated aliquot sharpen to a singular peak at 69 Ma in the treated aliquot. In the untreated aliquot, the 68-72 Ma age populations make up ~1.2% of the total grains in the untreated aliquot, whereas in the treated aliquot the 69 Ma peak age population represents ~1.3% of total grains. There is also an older shift from the 93 Ma peak in the untreated aliquot to ~98 Ma in the treated aliquot. However, due to the low number of grains within these peak age populations, it is possible these shifts are related to subtle differences between the two fractions and not a consequence of the chemical abrasion

process. For example, the peak height change in the 93 Ma population (3 grains; untreated aliquot) to the 98-100 Ma
(4 grains; treated aliquot) represents the addition of a single grain (Fig. 7c). Other shifts and changes in peak age
populations that are <120 Ma (Fig. 7c) cannot be confidently constrained due to the low number of analyses that define
those populations (1-2 grains). The fraction of concordant grains is indistinguishable between treated and untreated
aliquots of NM8A (Fig. 8). Quantitative comparisons of treated and untreated aliquots further support these results
and similarity, cross-correlation, and likeness values are all >0.8 (Fig. 6). The Similarity value of 0.957 indicates a
strong overlap in the number and proportion of detrital zircon peak age populations overall. Slightly lower, but still
high, cross-correlation (0.851) and likeness (0.816) coefficients support minor shifts in the number of peak age
populations (modes) and changes in peak heights (magnitude).


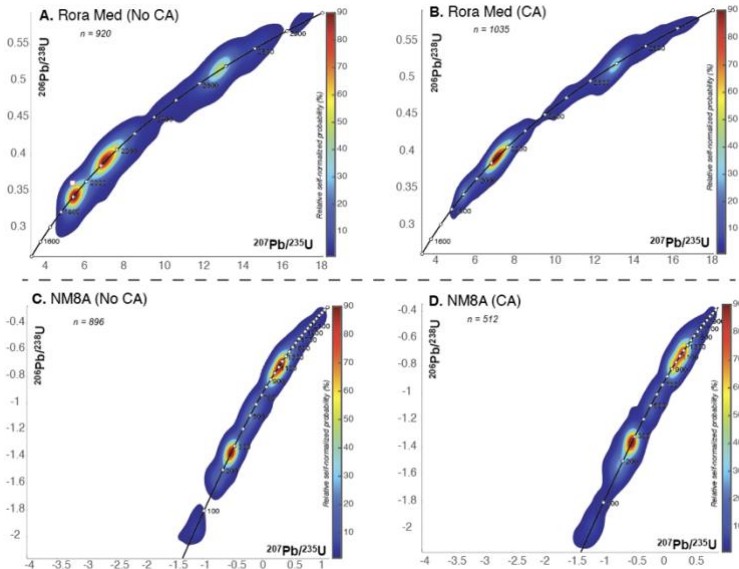

**Figure 8.** Density contour concordia diagrams for not chemically abraded and chemically abraded aliquots of detrital zircon samples NM8A and Rora Med (A-D). There is substantial improvement in the scatter of concordant analyses in the chemically abraded aliquot of the Proterozoic Rora Med sample compared to the untreated aliquot (A-B). However, both aliquots of the Phanerozoic NM8A sample are indistinguishable (C-D). Please note that the concordia diagrams for NM8A (C-D) are plotted on a logarithmic scale to best display

the large range Phanerozoic to Precambrian ages.

**4. Discussion**
Our study shows that chemical abrasion prior to LA-ICP-MS analysis does not negatively affect resulting U-
Pb dates provided chemically abraded reference materials are used as the primary reference material for calibration
(e.g., Crowley et al., 2014; von Quadt et al., 2014). We also show that chemical abrasion is extremely effective in
mitigating the effects of Pb-loss in LA-ICP-MS U-Pb dating of zircon that has experienced substantial radiation
damage. Significant improvement was observed in both $^{206}Pb/^{238}U$ and $^{207}Pb/^{206}Pb$ dates of MIGU-02 zircon relative
to ID-TIMS results, and also the efficiency of the analyses was dramatically improved by focusing LA-ICP-MS
analyses on only those grains/fragments that survived the chemical abrasion process and had not sustained significant
radiation damage. These results reinforce the observations of previous studies that used this approach (Crowley et al.,
2014; von Quadt et al., 2014) and suggested that the CA-LA-ICP-MS method can be valuable for studies that need
increased precision and accuracy in LA-ICP-MS U-Pb zircon analyses. Although, care must be taken to ensure data
is not biased when this pre-treatment is applied.
One important consideration is whether chemical abrasion negatively affects the laser ablation process.
Variations in laser ablation pit depths have been directly correlated to the density of the zircon matrix (Marillo-Sialer
et al., 2014; 2016) and changes in ablation rate change down-hole fractionation. Previous research has indicted that
annealing leads to lower ablation rates compared to unannealed zircon and more homogenous rates across annealed
zircon with variable initial degrees of radiation damage (Marillo-Sialer et al., 2016). These observations support results
from Campbell and Allen (2012) that thermal annealing zircon samples prior to LA-ICP-MS will reduce matrix-related
bias and improve accuracy and precision. However, the impact of CA on laser coupling and ablation rates is not as
well characterized. Crowley et al. (2014) found that treated zircons had pit depths that were 25% shallower than
untreated aliquots of zircons that experienced extensive Pb-loss. This is identical to the results for the metamict MIGU-
02 sample in this study in which chemically abraded zircon also had ~25% shallower ablation pits than untreated
zircon (Table S15). The shallower pit depths could be driven by less effective laser coupling in treated aliquots due to
small-scale etching and creation of a 3-D porous texture by partial dissolution (Crowley et al., 2014). In comparison,
laser ablation rates and pit depths in treated and untreated aliquots of primary reference materials were nearly identical
(Fig. 2). This supports conclusions drawn by Crowley et al. (2014) that the extent to which zircon is impacted by CA
is dependent on U concentration and the presence of physical defects and highlights the importance of incorporating
a wide range chemically abraded reference materials in each analytical session.
Our data, and the data of Crowley et al. (2014) and von Quadt et al. (2014), indicate that provided care is
taken to use chemically abraded reference materials, the pre-treatment will mitigates Pb-loss and lead to increased
accuracy in LA-ICP-MS U-Pb zircon analyses. Given this apparent benefit, it is natural to extend the technique to
detrital zircon and test the advantages and disadvantages afforded by this method. Crowley et al. (2014) first used this
approach on an Archean graywacke and showed that it did not significantly bias their results. However, this technique
has not been widely used over the last decade. We report similar results to previous studies in that chemical abrasion
does not significantly bias results or negatively affect LA-ICP-MS dates. This is supported by quantitative comparison
tests of the treated and untreated aliquots of both detrital zircon samples that show high degrees of similarity between
aliquots. Minor changes in cross-correlation and likeness values are a result of minor changes in the number of and
magnitude of specific peak age populations between treated and untreated aliquots (Fig. 6 and 7). Our results indicate
that a chemical abrasion pre-treatment may help resolve finer scale features in detrital zircon spectra from the Cenozoic
to the Archean. We attribute this increased resolution mainly to the mitigation of Pb-loss leading to increased accuracy
of the resulting LA-ICP-MS U-Pb dates.
The mitigation of Pb-loss is most clearly observed in the sharpening of Neoproterozoic through Cenozoic
age populations because $^{206}Pb/^{238}U$ dates provide the most precise estimate for zircon crystallization in this age range.
Pb-loss can significantly affect the accuracy of dates in this age range since Pb-loss trajectories closely follow
concordia and can result in dates that have apparent concordance despite being inaccurate. These effects can be seen
most clearly in sample NM8A where age peaks narrowed and became more defined (e.g., 250-300 Ma peak age
populations) following chemical abrasion and some peak age populations shifted to slightly older dates (Fig. 7).
Assuming that the zircons that form these populations cooled below the temperature at which radiation damage is
effectively annealed at a similar time, then U content can be used as a proxy for radiation damage (Nasdala et al.,
2005; Widmann et al., 2019; McKanna et al., 2023). This is clearly observed for the treated and untreated aliquots of
igneous sample MIGU-02, where the treated aliquot has substantially lower $^{238}$U beam intensities and increased
concordance and accuracy in $^{206}$Pb/$^{238}$U dates (Figs. 3, 4, and 5). However, the thermal history is not known *a priori*
for detrital zircon datasets, meaning this same assessment applied to NM8A and Rora Med is more uncertain.

524       To examine whether the reduced Pb-loss we observed in the chemically abraded aliquot reflects the selective

dissolution of zircon with radiation damage, we compared zircon $^{238}$U beam intensity from a particular age range (250-
320 Ma) in NM8A as a first-order approximation (Fig. 9). We assume that the populations in this range likely have
the same low-T history, although this assumption cannot be tested with our data. We also note that these populations
showed the most significant sharpening following chemical abrasion (Fig. 7B). Figure 9 shows that the average $^{238}$U
beam intensity of treated grains in this age range is similar, but there is a distinct proportion of treated grains that have
lower $^{238}$U intensities. This supports our interpretation that CA is likely mitigating Pb-loss by dissolving zircon with
high U concentrations and that this process can be observed by the changes in the number of peaks, their shape, and
their magnitude between treated and untreated aliquots. These changes are especially apparent when comparing the
shift and change in magnitude of the Rora Med 2665-2685 Ma (untreated aliquot) peak age to 2690-2715 Ma (treated
aliquot).

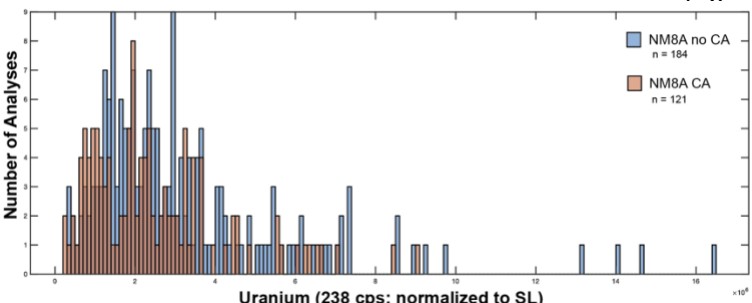


**Figure 9.** Histogram showing $^{238}$U cps (normalized to SL) for zircons in the peak age population between 250-320 Ma in detrital zircon sample NM8A. On the detrital zircon spectra, this age population narrows from one broad peak in the untreated aliquot to a well-defined, narrow

peak in the treated aliquot (Fig. 7b). Measured $^{238}$U cps from this peak age population of treated and untreated aliquots
are overall similar, but there is a distinguishable proportion of grains in the treated aliquot that have lower $^{238}$U cps.

546       Reduced Pb-loss in Mesoproterozoic and older zircon also benefits detrital zircon studies because ancient

Pb-loss can bias $^{207}$Pb/$^{206}$Pb dates of moderately discordant or even (analytically) concordant zircon toward
erroneously young values (Nemchin and Cawood, 2005). This effect has led many laboratories to filter for discordance
within their datasets. Thus, improving concordance will increase the proportion of dates that can be retained in a
detrital zircon study and improve confidence in the identification of peak age populations. One potential issue with
this approach is the possibility that entire zircon populations could be removed during chemical abrasion if they have
high degrees of radiation damage. Surprisingly, we did not see this effect in either NM8A nor Rora Med. This result
is surprising and may be sample specific, since Rora Med zircon from all age populations have low $^{238}$U beam
intensities (Fig. 10b). Although our RoraMed sample did not entirely lose any age populations during CA, changes in
the magnitude of peak heights suggests preferential dissolution of some age populations associated with Pb-loss. This
feature may be unique to Precambrian samples with overall low zircon U concentrations and/or recent exhumation of
the sedimentary rocks to the temperature conditions where radiation damage can accumulate and Pb-loss occurs.
Regardless, both NM8A and Rora Med have similarity values of >0.9 (Figs. 6 and &), indicating that similar peak age
populations and proportions are present in both treated and untreated aliquots. Therefore, it is unlikely that chemical
abrasion would impact detrital zircon spectra in a way that would make aliquots look as though they were sampling
different source terranes.

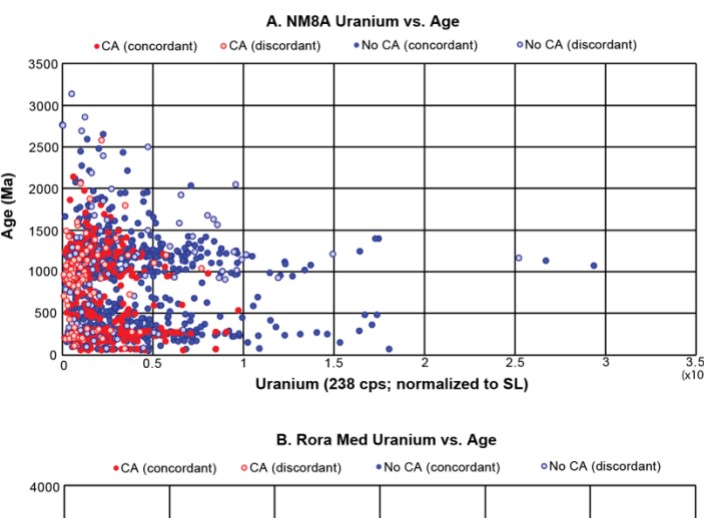

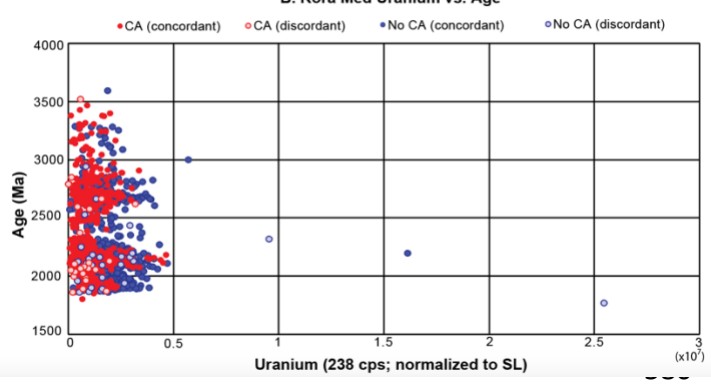

**Figure 10.** Scatter plot of $^{238}$U cps (normalized to SL) plotted against the age (Ma) for both treated and untreated aliquots of **A.** NM8A and **B.** Rora Med. Both concordant and discordant analyses are shown. Overall, CA appears effectively reduce scatter in $^{238}$U cps in all age populations compared to the untreated aliquot, but most significantly reduces scatter in Precambrian ages.


582       The nature of sediment transport may also work to remove metamict zircon prior to deposition in certain
environments. Hydraulic sorting, mechanical abrasion, and weathering, can naturally bias detrital zircon populations
present in a different lithologies (Malusa et al., 2013; Ibañez-Mejia et al., 2018). For example, Ewing et al. (2003)
noted that metamictization leads to structural damage of the zircon crystal structure and that this can be correlated to
a decrease in density and hardness. These changes lead metamict zircon to be more prone to destruction during river
transport (Fedo et al., 2003; Hay and Dempster, 2009a). In particular, Hay and Dempster (2009a) argue that inclusion-
rich and metamict zircon are broken during sediment transport, and that these fragments do not survive being
incorporated into clastic sandstone deposits. Instead, these smaller fragments can be swept out to more distal
depositional environments. Small zircon are also typically lost during sample preparation (Hietpas et al., 2011; Slama
and Kosler, 2012), meaning that both natural and laboratory processes may preferentially lead to a high proportion of
undamaged zircon in sandstone samples. Thus, while we did not observe the removal of specific age populations
following chemical abrasion in the two detrital zircon samples that were analyzed in this study, and there are reasons
to suspect that natural and laboratory processes will favor the analysis of undamaged zircon anyway, we recognize
that other samples may behave differently. Future users of this technique should carefully consider this possibility in
their datasets.
Another potential benefit of chemical abrasion is the preferential dissolution of inclusions within zircon
during the partial dissolution step (McKanna et al., 2023). Inclusions harbor $Pb_c$ that can be incorporated into the
analyzed volume during laser ablation, reducing the $Pb^*/Pb_c$ and limiting measurement precision and accuracy. When
comparing the $Pb^*/Pb_c$ ratios of treated and untreated aliquots of MIGU-02, we see a clear distinction that treated
zircons have a much higher $Pb^*/Pb_c$ ratio for similar ranges in $^{238}$U beam intensity (Fig. 11). We note that the overall
$^{238}$U beam intensities for the treated aliquot of MIGU-02 are low compared to the untreated aliquot, as we have already
shown that CA for metamict zircon effectively removes high-U zones where Pb-loss is most likely to have occurred
(see above discussion; Fig. 5). Regardless, the increased $Pb^*/Pb_c$ ratio for the treated aliquot of MIGU-02 shows that
this method is also efficient in removing inclusions with high $Pb_c$ content and/or highly damaged domains where $Pb_c$
might have been introduced by fluids. These two effects are correlated with an increased fraction of concordant grains
and increased precision and accuracy in $^{206}Pb/^{238}U$ zircon dates in the chemically abraded aliquot. These observations
support the benefits of utilizing CA prior to LA-ICP-MS measurements in metamict igneous zircon suites. A
comparison of the $Pb^*/Pb_c$ ratios in treated and untreated aliquots of detrital samples (Fig. S18) shows similar behavior
in Rora Med with slightly higher Pb*/Pbc in the treated aliquot and no change in the Pb*/Pbc ratios in NM8A. We
conclude that it is likely that CA is removing inclusions prior to analysis of detrital zircons as well. However, it is
difficult to isolate this effect since detrital zircons are sourced from various terranes and we cannot confidently
compare the Pb*/Pbc of zircon with the same age, U concentration, and thermal history.

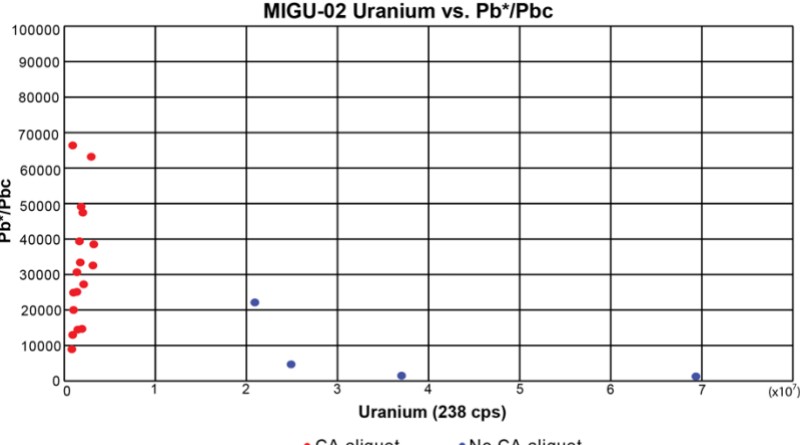

**Figure 11.** Pb*/Pbc ratios are plotted against $^{238}$U cps for MIGU-02. The Pb*/Pbc ratios in the treated aliquot of MIGU-02 are significantly higher than the untreated aliquot for similar concentrations of U. Higher Pb*/Pbc ratios in the treated aliquot of MIGU-02 can be attributed to reduction of Pbc by removal of inclusions.



### 6. Conclusions and Recommended Applications

Chemical abrasion is a widely used tool in the zircon U-Pb ID-TIMS community (*see reviews in* Schoene,
2014; Schaltegger et al., 2015), where it has been repeatedly shown to mitigate the negative effects on age accuracy
introduced by Pb-loss (Mundil et al., 2004; Mattinson, 2005; Widmann et al., 2019). Recent efforts to extend chemical
abrasion to LA-ICP-MS analyses have also shown that this pre-treatment can be beneficial (Crowley et al., 2014; Von

Quadt et al., 2014; McKanna et al., 2023; Sharman and Malkowski, 2023). The extension of this pre-treatment to large-n detrital zircon analyses is a natural outgrowth of these efforts. Our results indicate no negative effects from chemical abrasion prior to LA-ICP-MS analyses and that the technique results in improved percentages of concordant grains, reduced uncertainty, and, at least for the highly radiation damaged igneous sample we studied here, accuracy of measured U-Pb dates. For DZ samples, these benefits appear to translate to more defined and slightly older $^{206}Pb/^{238}U$ age peaks for Phanerozoic zircon, and more concordant analyses, and in some cases slightly older $^{207}Pb/^{206}Pb$ dates for Precambrian zircon. One potential drawback of this pre-treatment is the possibility that age populations characterized by high-U zircon may be selectively dissolved during chemical abrasion. We did not observe this effect in either of our tested samples. However, we remain wary of its possibility in other samples with highly damaged Precambrian zircon populations, and so future practitioners are advised caution. The differences between age distributions in our analyzed detrital zircon spectra are slight and indicate that the Pb-loss present in typical untreated analyses would not significantly alter the interpretation of sediment source terranes at a broad scale. However, chemical abrasion did sharpen several Phanerozoic peak ages and led to an increased percentage of concordant grains in Precambrian zircon populations. This indicates that the pre-treatment may be useful in certain scenarios in which researchers may require increased resolution of detrital zircon age spectra to distinguish fine-scale variations in provenance, sediment source terranes, or source characteristics. Ongoing research aims to test this method's impact on improving the precision and accuracy of maximum depositional age (MDA) estimations.

**Supplement**

All datasets utilized in this study are available in the Supplementary Materia online at:

**Author contribution**

EED, MPE, and MIM all helped design experiments for treated and untreated zircon aliquots and EED prepared all bulk chemically abraded aliquots and conducted CA-ID-TIMS experiments. FM and EED both conducted LA-ICP-MS and CA-LA-ICP-MS experiments. FM and MIM designed and FM conducted zircon optical profilometry experiments. All authors participated in the interpretation and discussion of results. EED prepared the figures and manuscript.

**Competing Interests**

The authors declare no competing interests.

**Acknowledgments**

We thank the Arizona LaserChron Center (ALC) for sharing samples and reference materials and for helping analyze these samples. Specifically, we thank G. Gehrels, M. Pecha, D. Alberts, W. Allen, M. Foley, and T. Milster. We also thank R. Ickert for help designing a system for bulk CA at Purdue. All LA-ICPMS measurements were made at the Arizona LaserChron Center under NSF-EAR 2050246 for support of the Arizona LaserChron Center and all CA steps and CA-ID-TIMS measurements were completed at Purdue University's Radiogenic Isotope Geology Lab (RIGL) under NSF-EAR-2151277 to M. Eddy.

670

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
