# Peer review of "Minimizing the effects of Pb-loss in detrital and igneous U-Pb zircon geochronology by CA-LA-ICP-MS"

_Geochronology, 2023_

## Referee Comment (RC4)

[revised manuscript text omitted]

Chart Title

Bias %

Bias %

| | non-CA bias diff run1-run2 | CA bias diff run1-run2 |
|---|---|---|
| FCT | 1.556 | 2.04 |
| GHR1 | -2.473 | 0.369 |
| 49217 | 3.574 | -0.338 |
| Ples | -1.589 | -1.522 |
| Temora 2 | 1.469 | 0.488 |
| R33 | -0.619 | -1.397 |
| SLM | -3.205 | 0.014 |
| SLF | 0.063 | 0.182 |
| 91500 | -1.251 | -1.541 |
| FC1 | 0.075 | -0.149 |
| Oracle | 0.07 | -1.261 |
| QGNG | 0.473 | -0.276 |
| OG1 | 1.673 | -0.015 |

Chart Title

Ave bias%

Review of Donaghy et al – Increased accuracy and precision in igneous and detrital zircon geochronology using CA-LA-ICPMS.

The authors look to demonstrate improved accuracy and precision of U-Pb data acquired utilising a chemical abrasion sample preparation method, compared to laser ablation analysis without using this. I have provided comments in the attached pdf, but some key points are:

1) Please state the uncertainty level in all figures (including supplementary).
2) Uncertainties should be quoted to 2 significant figures with ages/ratios/values quoted to the same number of decimal places as the uncertainty.
3) MSWD's should be quoted to 2 significant figures
4) Please provide a metadata table for the LA-ICP-MS work (and ideally ID-TIMS also).
5) What reference values were used for FC-1 as the primary reference material?
6) Imaging and targeting to avoid zonation/inclusions/xenocrysts, especially in the younger zircons, may have avoided some of the issues discussed. For balance, the usefulness of imaging for this purpose could be mentioned.
7) Resolution of concordance is mentioned a lot with CA stated as improving concordance and resolution of concordance. However, it is equally stated that the same accuracy is achieved between CA'd and non-CA'd datasets. Illustration of this discussion would be much easier if the bias of the reference materials was tabulated in the manuscript since this is the fundamental premise of the paper. Taking the biases quoted in the supplementary plots, it can be seen in the figures below that the bias and pattern of both treated and untreated runs are equivalent.

[Figure]

[Figure]

Note the negative bias for 91500 might be accounted for by using CA'd reference values from (Schoene et al 2006 or Horstwood et al 2016) rather than the non-CA values (Wiedenbeck et al 1995).

These plots highlight the reduced scatter in the CA data whilst not changing the overall bias. Plotting the data another way, as the average bias between runs 1 & 2 for CA'd and non-CA'd aliquots, highlights the similarity in bias between the two data sets. In this example then, CA doesn't appear to improve concordancy (since the bias isnt changing at the +/-1% level), but is improving scatter.

[Figure]

In this respect, CA can probably be said to be improving the resolution of concordancy, however, stated in the way of the manuscript, this assertion is unquantified. The language around 'improved resolution and precision' is very loose and should be tightened with better quantification using the data acquired. This would be aided by being able to link the analyses to the nature of the material targeted but I appreciate this wasn't the approach taken. However, when looking at 'resolution' of detrital zircon spectra, knowing that some of the many analyses were not straddling age zones would be important, so that the shape and resolution of the age peak can be quantified and compared with another, by reducing the 'baseline' of potentially mixed measurements. In the absence of being able to do this, perhaps a different form of words or better explanation of some of the constraints on peak shape and dispersion might be useful, followed by described quantification of these. The outcome would be more supportive of the authors arguments for improvements resulting from CA-LA-ICPMS.

---

## Author Response (AR1)

**Erin E. Donaghy**

PhD Candidate | edonaghy@purdue.edu | (609) 510-1980 | https://sites.google.com/view/erinedonaghy

EAPS Department | 550 Stadium Mall Dr. | Purdue University | West Lafayette, IN 47907

To: Editors of *Geochronology*                                           January 24th, 2024

Dear Editors,

Enclosed is the revised manuscript "*Minimizing the effects of Pb-loss in detrital and igneous U-Pb zircon geochronology by CA-LA-ICP-MS*" for publication in *Geochronology* (GCHRON-2023-20).

This manuscript provides a detailed study quantifying the effect of chemical abrasion (CA) on detrital and igneous zircons prior to laser ablation-inductively coupled plasma mass spectrometry (LA-ICP-MS). We thank the reviewers for their thoughtful reviews and comments and have made changes to the original manuscript accordingly. Reviewers for this manuscript were David Chew, Marcel Guillong, and Matthew Horstwood. Overall, the primary theme of the suggested reviews centered around 1) Language used around defining how accuracy and precision was improved by using CA, 2) How uranium concentrations were calculated for CA-LA-ICP-MS and LA-ICP-MS analyses, and 3) Provide a more detailed quantitative comparison of treated and untreated detrital zircon aliquots. Line numbers in this document refer to the line numbers that are in the final revised version of the track-changes manuscript. We addressed these reviews in a robust and extremely thorough manner by collecting and adding additional datasets (optical profilometry of zircons) and by modifying our discussion based on new Uranium value calculations. These changes to the original manuscript will be detailed below in a point-by-point response to reviewers.

**Revisions Related to Similar Comments by All Reviewers**

**Lines 1-2**: As recommended by all reviewers, we changed the title of the manuscript from "Increased accuracy and precision in igneous and detrital zircon geochronology using CA-LA-ICP-MS to "Minimizing the effects of Pb-loss in detrital and igneous U-Pb zircon geochronology by CA-LA-ICP-MS". We believe this change better reflects the key findings of the study. Improved accuracy has been observed in analyses of chemically abraded (CA) igneous zircon (this study, von Quadt et al., 2014). Demonstrating this same improvement for detrital zircon is much more difficult given that we cannot re-date every analyzed zircon by a second method, like ID-TIMS. Instead, we have to infer that the detrital zircon will respond to chemical abrasion in the same manner as the igneous zircon and minimize the effects of Pb-loss. This should improve accuracy in detrital zircon measurements and lead to predictable changes in DZ spectra. Namely that age populations will sharpen and move toward slightly older dates as Pb-loss is mitigated. Zircon populations with high degrees of radiation damage may also be preferentially dissolved in the process and lead to changes in the relative importance of each age population. We did note some of these changes in the manuscript.

**Lines 71-84:** We expanded the introduction to address low-T Pb-loss in detrital zircons, as recommended by reviewers and public comment by Trystan Herriott.

**How was Uranium concentration quantified:** Uranium concentrations, as reported in our original submission, were semiquantitative, calculated using simple standard-sample bracketing relative to the average U concentrations for our primary reference material. Quantification of trace element concentrations using LA-ICPMS are rigorously done using internal normalization relative to a stoichiometric element (e.g., Zr or Si), but we were not able to do this with our method. This was because measuring isotopes in the Zr or Si mass range would have required a magnet jump with the Element2, which would significantly slow down our analyses. The reviewers are correct that this should be better explained, so in our revised manuscript we have included additional clarifications about our analytical method and the reason why quantification was not done using an internal standard (**Lines 186-198**). Furthermore, because the U concentrations we reported in our original manuscript were not strictly quantitative, and the CA treatment of reference materials has the potential to skew the semiquantitative calculations originally performed, we have modified our manuscript to avoid the use of U concentrations in the text and figures. We now base our observations on the $^{238}U$ cps for each analysis, reported after performing a simple inter-session normalization for instrumental sensitivity. We explain this procedure in greater detail in the revised manuscript (**Lines 186-198**), but in brief: we note that the $^{238}U$ cps of our SL crystal were very homogeneous between and within runs of treated and untreated aliquots, so we used these as reference to normalize the cps of $^{238}U$ for all sessions. By removing minor variations in sensitivity using this simple approach, we now focus our discussion on the effects of chemical abrasion as a function of $^{238}U$ cps rather than U concentration. While this approach does not affect our general conclusions, it does resolve two key issues: i) removes the need to build our discussion around U concentrations, as these were not determined quantitatively; ii) removes possible inaccuracies introduced by the effects that chemical abrasion of reference materials can have on U (semiquantitative) concentrations calculated by simple standard-sample bracketing.

Discussions on how U concentrations varied in reference materials, igneous MIGU-02 sample, and the detrital zircon samples (NM8A and Rora Med) were all modified to $^{238}U$ cps. New supplementary figures were created (**Supplementary Figures S14-S17)** to show the $^{238}U$ cps in all runs for the reference materials in the round robin, igneous MIGU sample, and the detrital zircon samples. These figures show that SLF has an overall homogeneous $^{238}U$ cps within and across different runs of treated and untreated aliquots of reference materials and how all samples were normalized to the average treated or untreated SLF $^{238}U$ cps, respectively.

In the Discussion section, we changed any language from U concentration (ppm) to 238U cps. Discussion of how 238U cps varies between treated and untreated detrital zircon aliquots is documented in **Lines 1055-1260.**

Additionally, **Figure 5, Figure 9, Figure 10, and Figure 11** were all modified and redrafted to plot using $^{238}U$ cps normalized the SLF. The **Supplementary Figure S18** also has been included to show all analyses from treated and untreated detrital zircon runs plotted as $^{238}U$ cps versus

Pbc/Pb*, to aid in the discussion on how CA improves this ratio by removing zones of high U and Pbc.

**How does CA impact laser coupling and laser ablation rates:** New zircon optical profilometry data was collected to quantify the depth and shape of laser ablation pits in treated and untreated grains of primary reference materials and igneous sample MIGU-02. This will help us understand how CA influences laser coupling and ablations rates. Although zircon mounts are polished, Crowley et al. (2014) and McKanna et al. (2023) shows that chemical etching and 3-D porous textures can occur throughout the zircon crystal interior. The methods for zircon optical profilometry completed in this study are added to the Methods section (Now **2.3. Zircon Optical Profilometry)** with the methods detailed between **Lines 286-294.** A discussion on the effects of CA on laser ablation coupling and behavior was also added in **Lines 455-473.** This discussion outlines how our zircon profilometry experiments were set up to calculate laser ablation rates and results.

The discussion on how laser ablation rates were calculated using reference materials is between **lines 465-473.** Pit depth variations for treated and untreated aliquots of MIGU-02 are discussed in **Lines 700-704.** We added an extensive discussion on whether or not chemical abrasion negatively impacts the laser ablation process to the Discussion **Lines 951-966.** This discussion addresses concerns from reviewers on how pit depths and matrix-related effects could possibly change between treated and untreated aliquots.

We included new **Supplementary Tables S14 and S15** with pit depth data for primary reference materials and MIGU-02. We also created a new figure (**Figure 2)** that shows how laser ablation rates were calculated for treated and untreated aliquots and an example of one of our VEECO surface data maps. Due to the high volume of VEECE surface maps and crosscut profiles produced while collecting this new dataset, additional images and profiles will be available upon request. All critical data that was collected from estimating pit depths from these images and profiles is summarized in **Supplementary Tables S14 and S15.** Because of the addition of the new **Figure 2**, all following Figures were modified (Figure 2 in the original manuscript now becomes Figure 3, Figure 3 becomes Figure 4, and so on). There is now a total of **eleven** figures in the revised manuscript. All Figures in this document refer to the Figure number as stated in the revised manuscript.

**Quantitative Comparison of Detrital Zircon Samples Rora Med and NM8A:** All reviewers suggested a more detailed quantitative comparison of treated and untreated aliquots of detrital zircon samples Rora Med and NM8A. We used Saylor and Sundell (2016) DZStats software to complete a quantitative comparison of treated and untreated aliquots of both Rora Med and NM8A. The methods and details of the tests used in DZStats is detailed in **Lines 199-283.** Results of DZStats for Rora med are in **Lines 816-820** and for NM8A in **Lines 900-904).** Additionally, we also calculated the proportion of zircons (out of the total) making up peak age populations to compare how this varied between treated and untreated aliquots. This was incorporated into the discussion (**Lines 808-816, 878-897).**

We modified **Figures 6 and 7** to show the proportion of grains (out of the total) comprising each peak in treated and untreated aliquots to allow for a more robust discussion on how those proportions were changing after CA. Additionally, we utilized DZStats software by Saylor and

Sundell (2016) to calculate the similarity, cross-correlation, and likeness values and incorporated this comparison into our discussion.

In regard to **David Chew's comment on CA selectively dissolving high U grains with reference to the 1890 peak age population in Rora Med:** To address this comment, we wanted to better quantify the decrease in peak height from the treated to the untreated aliquot and compare U concentrations associated with this peak age population. The total percent of zircons making up the 1890 peak age population in the untreated aliquot of Rora Med is ~7.4% compared to a decrease to ~3% of total grains in the treated aliquot. This decrease correlates to the observed change in peak height on Fig. 5. To determine if CA preferentially dissolved zircons of high U associated with this peak age population in the treated aliquot, in our revised manuscript we now compare the 238U cps between treated and untreated aliquots (discussed above). We updated **Figure 10** to compare the 238U cps between treated and untreated aliquots of Rora Med (and NM8A). In the updated version, the treated aliquot of Rora Med has overall lower 238U cps values compared to the untreated aliquot of Rora Med. This difference is especially noticeable for the 1800-2000 Ma and the 2600-2800 Ma age populations. For example, in the untreated aliquot, there are 70 grains associated with the 1880-1900 Ma age population. About 33% of the grains between 1880-1900 Ma make up the highest 238U cps values (238U cps > 2,171,019) for the entire aliquot, but these grains only represent ~2.5% of the total aliquot (23/920 grains). In contrast, there are a total of 33 grains in the treated aliquot for this age population, and ~42% of zircons (14 total grains) in the 1880-1900 Ma age population make up the highest 238U cps values (238U cps > 1,355,272). This represents ~3% of all zircons in the treated aliquot (14/1035 grains). Overall, there are significantly less zircon grains of the 1880-1900 Ma age population in the treated aliquot, and the overall 238U cps values are lower. This difference is likely due to mitigation of Pb-loss in the treated aliquot, but overall, the range of U values associated with the 1890 peak age population in both treated and untreated aliquots suggests that CA does not selectively dissolve only high U grains.

**Figures 6 and 7:** We have added the proportion of analyzed grains (out of the total) that comprise the age populations in peaks. We also added the DZStats (Saylor and Sundell, 2016) similarity, likeness, and cross-correlation values to each of the probability density plots (PDPs). All statistical quantification tests are summarized and reported in **Supplementary Table 22.**

_**Response to Marcel Guillong Review**_
**In reference to the comment: "the accuracy and precision of materials analyzed using CA and non-CA seems not improved…".** We have also gone back through in the revised manuscript to clarify language in Section 3.1. Here, the reviewer's main concern is that accuracy and precision of reference materials was not improved between treated and untreated aliquots. However, we note that we did not necessarily anticipate this would be the case, because reference materials used for U-Pb geochronology are dominantly concordant (i.e., Pb loss is rare or absent) and 'well-behaved', and these are reasons why they are chosen as reference materials in the first place. Instead, the main objective of analyzing all these reference materials after performing chemical abrasion was to demonstrate that the accuracy and precision of our U-Pb data would not be negatively affected, and hence that the dates from our unknows are reliable over a wide age range, We did, however, note some slight improvement in the U-Pb systematics of some reference materials, given that fewer analyses were discarded due to discordance. However, these materials were Proterozoic in age and are primarily used for their homogenous $^{207}Pb/^{206}Pb$ which is less sensitive to recent Pbloss. Overall, the behavior between treated and untreated reference materials is similar, and thus the objective of these analyses (i.e., demonstrate our method does not negatively affect accuracy) was met. Again, this result is expected because reference materials are selected due to homogenous isotopic compositions and excellent behavior during analysis. For the purposes of this manuscript, these results demonstrate that chemical abrasion does not systematically bias our U-Pb results.

**In reference to the comment: "Comparison for MIGU-02 is not entirely fair as for the CA about 150 zircons were used vs non-CA only 35 grains being used…":** We acknowledge that there is a balance to be struck when deciding whether to utilize chemical abrasion prior to LA-ICP-MS analyses. For samples with significant radiation damage, there is always the possibility that the entire sample will dissolve. Running a high-n on highly damaged zircon might ultimately yield enough concordant analyses to make a confident age determination. However, the concordant analyses for our metamict igneous sample MIGU-02 were inaccurate by up to -11% for the $^{207}Pb/^{206}Pb$ dates and -21% for the $^{206}Pb/^{238}U$ dates. The chemically abraded aliquot didn't have these issues. For many felsic igneous samples zircon yields are not an issue, so performing the CA treatment on many crystals–even if the majority dissolve–as a means to optimize analytical time on the LA-ICPMS and enhance accuracy is, in our experience, a worthwhile approach. We think that chemical abrasion's efficacy at reducing Pb-loss, its relative ease and low cost in the laboratory, and the possibility of optimizing 'beam time' by only focusing on those grains that will yield concordant results, make it a worthwhile step in U-Pb zircon analyses by LA-ICP-MS. However, we will better acknowledge the potential drawbacks for samples with high degrees of radiation damage in the revised manuscript (**Lines 704-710**).

**In reference to comment: "How CA may impact laser coupling with the zircon…"**. We collected new optical profilometry data on treated and untreated aliquots of primary reference materials and igneous sample MIGU-02. This data allows us to quantify the depth and shape of laser ablation pits in treated and untreated grains and can help us understand how CA influences laser coupling and ablations rates. Although zircon mounts are polished, Crowley et al. (2014) and McKanna et al. (2023) shows that chemical etching and 3-D porous textures can occur throughout the zircon crystal interior. Discussion on matrix effects is in **Lines 455-473** and the methods for zircon profilometry is outlined between **Lines 285-294.**

We included new **Supplementary Tables S14 and S15** with pit depth data for primary reference materials and MIGU-02. We also created a new figure (**Figure 2)** that shows how laser ablation rates were calculated for treated and untreated aliquots and an example of one of our VEECO surface data maps. Due to the high volume of VEECE surface maps and crosscut profiles produced while collecting this new dataset, additional images and profiles will be available upon request. All critical data that was collected from estimating pit depths from these images and profiles is summarized in **Supplementary Tables S14 and S15.**

**As recommended by Reviewer Guillong**, we changed the word 'standard' to **'reference material'** throughout the entirety of the text.

**In reference to the comment: "Rank order plots are not ranked"**. We went back and replotted all rank order plots and ranked them by age (youngest to oldest) in **Figure 3** and all supplementary rank order plots for the reference materials (**Supplementary Figures S1-S13).**

**In reference to the comment to discuss how trace element (TE) concentrations respond to CA**: This is outside the scope of our study, and we implemented no changes here as we don't have the necessary data to address it.

**Figure 4A:** We changed the scaling to better display the ellipses. If we were to plot the discarded analyses, then no detail would be observed, making it difficult to see any detail in concordant analyses.

*Response to David Chew Review*
**In reference to better defining the methods used in the round robin:** We added more language in the methods section for outlining the term 'round robin' in context of analysis setup for this study, **Lines 175-185.**

**In reference to the comment on if zircon aliquots must be pure for bulk CA:** In this study, the separates that were bulk chemically abraded were 100% zircon. However, author Donaghy has implemented this method on detrital zircon samples for other research and has used separates that were ~85-90% zircon. The acid dissolution does effectively remove the other mineral phases and leaves a sample that is 100% zircon. However, this research is not published or reported yet.

**In reference to the comment on the origin of the 91500 crystal:** the aliquot we used was obtained from the International Association of Geoanalysts (https://iageo.com/zircon-91500/) so we believe its origin is robust. That said, in past research, the 91500 reference material has shown substantial negative age offset (Gehrels et al., 2008; Schoene et al., 2014), but the origin of these offsets has remained enigmatic. So, while the offset in this study is not entirely surprising, we expanded our discussion to briefly highlight this on our revised manuscript (**Lines 430-443).** We also expand this discussion to address matrix effects, which can also cause age offset. This discussion is located in **Lines 455-473.**

**In reference to adding fluence on each sample**: We added **Supplementary Table S23** which documents the laser and mass spectrometry parameters and metadata.

**Figure 3:** Reviewer caught a type-o in the n-value listed in Figure 3 compared to the correct value written in line 221. Figure 2 listed the wrong total number of analyses (n=35 versus n=20). Figure 3 was changed in the revised figure and no other changes were necessary as calculations were based off a n=20 value.

*Response to Matthew Horstwood Review*
**In response to stating the uncertainty level in all figures:** We went back and made sure to state that the analyses are reported in 2-sigma uncertainty for all Figures and Supplementary Figures.

**In response to uncertainties and MSWDs, age ratios, and uncertainties being quoted to 2 significant figures:** We went back and modified all Figures (1, 3) and Supplementary Figures (S1-S13) to display these values with 2 significant figures. They are also reported with 2 significant figures in the text **Lines 510-525.**

**In response to metadata tables for LA-ICP-MS:** We included a new **Supplementary Table** with metadata from the Arizona LaserChron (ALC) mass spectrometer.

**In response to the recommended figure tabulating % bias of treated and untreated aliquots**. We did not include an additional figure plotting the % bias of the treated and untreated aliquots of reference materials to compare. We believe that **Figure 1** is adequate in showing the age offset and the overall reduced scatter in the chemically abraded aliquots. However, we did tighten up vague language surround statements such as 'improved resolution and precision' and CA 'improving the resolution of concordancy'. In the **3. Results** section, we revised language to reflect the change in percentage of **retained** concordant grains versus referring to 'improved concordancy'(**Lines 341-343**). Instead of referring to 'improving precision and accuracy, we instead discussed how the **scatter** was improved from untreated to treated analyses (**Lines 349-354).**

Thank you for your help in the revision of and publication of this manuscript. We note that the figures in the merged PDF file are reduced in quality. We have original figure files in PDF and .ai format for higher resolution figures, if needed for final typesetting and formatting of the manuscript. Please address correspondence regarding the manuscript to Erin Donaghy by telephone at (609)-510-1980 or email (edonaghy@purdue.edu).

Best Regards,

Erin Donaghy, Dr. Michael Eddy, Dr. Mauricio Ibañez-Mejia, and Dr. Federico Moreno